# Microangiopathy in Ocular Sarcoidosis Using Fluorescein Gonio and Fundus Angiography from Diagnostic and Therapeutic Aspects

**DOI:** 10.3390/diagnostics11010039

**Published:** 2020-12-28

**Authors:** Teruhiko Hamanaka, Noriko Akabane, Tetsuro Sakurai, Soichiro Ikushima, Toshio Kumasaka, Tamiko Takemura

**Affiliations:** 1Department of Ophthalmology, Japanese Red Cross Medical Center, Tokyo 150-8935, Japan; 2Department of Ophthalmology, Toho University Ohashi Medical Center, Tokyo 153-8515, Japan; nonakabane@hotmail.com; 3Akabane Eye Clinic, Takasaki 370-0801, Japan; 4School of General and Management Studies, Suwa University of Science, Suwa 391-0292, Japan; s15002@gug.math.chuo-u.ac.jp; 5Department of Internal Medicine, Japanese Red Cross Medical Center, Tokyo 150-8935, Japan; s-ikushima@ab.auone-net.jp; 6Department of Pathology, Japanese Red Cross Medical Center, Tokyo 150-8935, Japan; kumasaka_toshio@med.jrc.or.jp; 7Department of Pathology, Kanagawa Cardiovascular and Respiratory Center, Yokohama 236-0051, Japan; tamikobyori@gmail.com

**Keywords:** angle-neovascularization, sarcoidosis, glaucoma, microangiopathy, endothelial damage, trabeculectomy, tube-shunt surgery, pathology

## Abstract

In this retrospective study, we investigated vascular abnormalities in sarcoidosis using fluorescein gonioangiography (FGA) to detect angle neovascularization (ANV), fundus fluorescein angiography (FFA), and pathological specimens from the aspects of microangiopathy. In 57 sarcoidosis patients, clinical data was reviewed by dividing the cases into three groups (Group I: histologically diagnosed; Group II: positive bilateral hilar lymphadenopathy (BHL); Group III: negative BHL). The FFA, FGA, and pathological examination data in the autopsy eyes and trabeculectomy specimens were investigated. FGA and FFA detected ANV (91%) and nodule-associated abnormalities (87%), respectively. No intraocular pressure (IOP) elevation was observed after continuous topical betamethasone, except in the steroid responder group. Maximum IOP had significant correlation with nodules in the angle (*p* = 0.02696) and visual field defect (*p* = 0.0151). Granulomas adjacent to blood vessels, including the Schlemm’s canal, and thickening of the retinal blood vessel wall caused occlusion of those vessels. Photocoagulation was required for retinal tears (14%) and the retinal blood vessel occlusion (7%). Suppression of IOP elevation via continuous topical betamethasone may be important to avoid irreversible outflow-route changes and optic-nerve damage, and the concept of microangiopathy in ocular sarcoidosis may be important for understanding the proper treatment of serious complications.

## 1. Introduction

The reported complications of ocular sarcoidosis include cataract, macular edema (ME), glaucoma, vitreous hemorrhage, retinal ischemia, retinal detachment, subretinal neovascularization (NV), and optic nerve disease [1,2,3,4,5]. Of those, glaucoma, cystoid ME, and retinal detachment are the most serious, as they can lead to blindness. Subjects sometimes become self-aware of ME when it occurs, however, subjects with retinal detachment and glaucoma tend to delay presentation at a medical facility for treatment. That delay in the initiation of treatment is reportedly one of the most strongly correlated factors associated with the lack of visual acuity (VA) improvement in cases of sarcoidosis [1]. Retinal tears usually occur in the extreme peripheral area of the retina, and most subjects are not aware of the tear until retinal detachment progresses to near the center of the visual field. Ocular hypertension due to sarcoidosis often accompanies recurrence of anterior chamber inflammation with mutton-fat-like keratic precipitates (KPs). However, the ocular signs are sometimes asymptomatic and the patients are not aware of ocular hypertension. Due to the silent and recurrent nature of sarcoidosis, ocular hypertension can often result in advanced-stage glaucoma. In a previous study by Obenauf et al. [6], it was reported that 10% of patients with iridocyclitis due to sarcoidosis have glaucoma, and that it sometimes develops into refractory glaucoma requiring filtration surgery. Those findings suggest that it may be important to make a diagnosis of ocular sarcoidosis at the early stage of the disease in order to initiate medication or proper treatment to prevent serious complications. Moreover, in that study, the authors reported a high frequency of involvement of the anterior segment in 84.7% of the cases, and in a previous study by Kawaguchi et al. [7], it was reported that nodules in the trabecular meshwork (TM) and/or tent-shaped peripheral anterior synechiae (PAS) had the highest values for all factors, including sensitivity and specificity. Karma and Laatikainen [8] were the first to introduce fluorescein angiography into the anterior segment using patients with sarcoidosis [8], and in that study, the authors reported that granulomas on the pupil were accompanied by NV. In the pathological examination of the aqueous outflow routes obtained from trabeculectomy (TRAB) in patients with glaucoma due to sarcoidosis, angle NV (ANV) was reportedly often observed in the TM [9]. 

To the best of our knowledge, this current study is the first to apply fluorescein gonioangiography (FGA) in patients with sarcoidosis. In addition, we used fundus fluorescein angiography (FFA) for the diagnosis of ocular sarcoidosis in our patients, as FFA is reportedly useful to detect abnormalities of the retina and choroid in patients afflicted with the disease [10,11,12]. It should be noted that Mikami et al. [13] proposed a new concept of microangiopathy in sarcoidosis from the histological observation in various organs. Therefore, we theorized that it may be interesting to investigate the clinical changes in ocular sarcoidosis from the aspect of microangiopathy using FGA and FFA, so we also investigated the pathological changes of the eyes with sarcoidosis correlated with the clinical findings of slit-lamp examination, FGA, and FFA using TRAB specimens and autopsy eyes from the aspects of microangiopathy.

## 2. Materials and Methods

This study was approved by the Institutional Review Board of the Japanese Red Cross Medical Center, Tokyo, Japan. In this study, we retrospectively reviewed the medical-record data of 57 Japanese patients who visited the Department of Ophthalmology at the Japanese Red Cross Medical Center with suspected sarcoidosis (Table 1A), patients who had undergone TRAB, and autopsy cases (Table 1B), and in whom it was possible to perform FGA and FFA with granulomatous inflammation in the anterior chamber and elevated intraocular pressure (IOP) accompanying nodules or PAS in the angle. The methods used for the FGA and FFA examinations and for obtaining the FGA and panoramic FFA photographs are as previously described [14]. For FGA and FFA, the worse eye of the patient, i.e., the eye with the higher IOP and that was more affected with ocular sarcoidosis, in principle, was chosen. In six patients with chorioretinitis or choroidal granuloma, indocyanine green angiography (IGA) was applied. All data of the best-corrected VA (BCVA), slit-lamp findings, fundus examination, and visual field defect (VFD) were from the eyes that received FGA and FFA. The BCVA findings were converted from decimal VA to logarithm of the minimum angle of resolution (logMAR). The data of retinal photocoagulation and cataract surgeries were from both eyes. The period in which the FGA and FFA examinations were performed was from September 1997 to October 2011, and the observation periods of those patients were from September 1997 to August 2020. The patient characteristics used for the clinical and pathological studies are listed in Table 1A,B, respectively, and those patients were divided into the following three groups according to the diagnosis criteria of ocular sarcoidosis [15]: Group I: definite ocular sarcoidosis (i.e., a biopsy-supported diagnosis with compatible uveitis), Group II: presumed ocular sarcoidosis (i.e., the presence of bilateral hilar lymphadenopathy (BHL) with compatible uveitis if a biopsy was not performed), Group III: probable or possible sarcoidosis (i.e., negative for BHL, yet with 3 suggestive intraocular signs). In the clinical study, although FGA was performed in all patients, it was possible to perform FFA in only 46 of the 57 patients.

In this study, the intraocular manifestations of ocular sarcoidosis were (1) KPs or iris nodules, (2) nodules in the TM or PAS, (3) snowballs or strings of pearls in the vitreous, (4) nodular or segmental periphlebitis and/or retinal macroaneurysm, (5) multiple chorioretinal peripheral lesions, (6) optic-disc nodule, and (7) bilaterality [15]. The FGA and FFA data were obtained by use of a slit lamp (SL-7F; Topcon Corporation, Tokyo, Japan) and a fundus camera (TRC-50VT; Topcon), respectively [14]. In some patients, the FGA images were obtained via the use of a fundus camera (TRC-50LX, TRC-50DX; Topcon). The definition of ANV in TM was as follows: arising perpendicularly from the iris root, and then turning to the different direction of running circumferentially or irregularly in the TM (Figure 1A). The tiny or budding NV that shows as dot staining of the iris root and gradual increase of the fluorescein leakage also included ANV (Figure 2). In the 46 patients in whom FFA was performed, panoramic FFA photographs were made in order to evaluate the diseased regions from the posterior pole to the extreme peripheral area. The location of the abnormal regions was determined using those panoramic photographs mounted by transparent sheets where radial peripapillary capillaries (RPC) and temporal raphe, mid periphery (within 8.2 optic-disc diameters), and extremely periphery (beyond 8.2 optic-disc diameters) were written (Figure 1B and Figure 2C) [14].

The locations of the abnormal findings were divided into the following three zones: Zone I: within RPC and temporal raphe, Zone II: within the mid periphery, yet outside of Zone I, and Zone III: outside of Zone II. The abnormal findings in the panoramic FFA images were granuloma in the retina, segmental vasculitis, perivascular nodule (PVN), diffuse retinal vasculitis, chorioretinal nodule, and diffuse chorioretinitis including chorioretinal atrophy. VFD was evaluated by Aulhorn-Greve (A-G) classification [16]. To investigate the reason for the complication of sarcoidosis, pathological examination was performed in 5 autopsy cases of sarcoidosis, 1 vitrectomy sample, and 14 TRAB specimens with secondary glaucoma due to sarcoidosis using light, as well as transmission and scanning electron microscopy. All patients who underwent TRAB were included in the clinical part of this study.

## 3. Results

The clinical part of the study involved 57 patients; 24 patients in Group I, 19 patients in Group II, and 14 patients in Group III. Of those 57 patients, 16 (67%) in Group I were diagnosed from lung tissues, yet two (8%) and six (25%) were diagnosed from skin biopsy and TRAB specimens, respectively. In those six patients who had granuloma in the TRAB specimens, two had been classified as Group II and four had been classified as Group III prior to granuloma being found in the specimens. Among the three Groups, no significant difference in patient age, gender, and observation period was observed (Table 2), and no significant difference in BCVA, max IOP, and VFD (A-G classification) was observed in the FGA and FFA examined eyes (Table 3). 

Most of the patients visited our clinic due to blurred vision or after being referred from another facility with suspected sarcoidosis due to the findings of a chest X-ray examination. There were three eyes with a BCVA of more than 0.4 logMAR units. Of those three eyes, two were end-stage glaucoma and one had a very small pupil with posterior synechia and cataract. In 12 patients (21%) who had an IOP of more than 40 mmHg, there was no complaint of eye pain, except for one patient who complained of a sense of tension in one eye, and the common complaint in those 12 patients was blurred vision. A significant correlation was found between the IOP and nodule in the angle (*p* = 0.02696) among the anterior-segment manifestations (KPs, nodule in the angle and PAS including vitreous) (Table 4). Ocular manifestations included inflammation in the anterior segment, vitreous, and posterior part, as mentioned above. Fine dust-like vitreous opacities were found in all patient eyes. Among 25 eyes (44%) with VFD, four showed no compatible changes with cup/disk ratio. However, in one VFD eye, glaucomatous cupping was not observed at the initial visit, yet became clear 13-years later. The remaining 21 eyes (37%) were diagnosed as secondary glaucoma due to sarcoidosis. PAS in all eyes was less than 25%, except for one eye with 50% PAS. In this study, logistic regression analysis was used to investigate whether the VFD and glaucoma surgery in our patients had any correlation with other parameters, and those findings revealed that maximum IOP (*p* = 0.0151) was the only significant correlation with VFD in A-G classification among patient age, gender, the location of the number of abnormal findings in the different FFA zones, the anterior segment abnormalities observed via slit lamp, and snowball opacity (Table 5A). Using the same method, the statistical correlation was investigated in the glaucoma surgery, the laser photocoagulation in the patients with a retinal tear, and the laser photocoagulation with retinal vein occlusion (RVO). Those findings revealed that glaucoma surgery had a significant correlation with Zone III in FFA and VFD in the A-G classification (Table 5B). However, no significant correlations were found in relation to laser photocoagulation.

In the 57 eyes, FGA examination was possible, but in 11 eyes, FFA data could not be obtained due to a small pupil caused by anterior synechia, vitreous opacity, or an inability to take panoramic photographs. FGA detected ANV in 52 (91%) of the 57 eyes, and the coexistence of a nodule on the TM in most of those 52 eyes (Figure 1A, inset in A). The use of a fundus camera made it possible to observe ANV by FGA (Figure 1C), and the gonioscopy examination findings often revealed bleeding in the TM (Figure 2A, inset). Six eyes (12%) showed positive for ANV by FGA, yet no visible granuloma on the TM (Figure 1C inset, and Figure 3A,D). Circumferential areas of ANV showed fluorescein leakage prior to the initiation of topical betamethasone administration (Figure 3B). However, at 1-week post initiation of the topical betamethasone administration, the leakage in those circumferential areas disappeared (Figure 3C). Post the administration of the topical betamethasone treatment, the granulomas disappeared (Figure 3D) and IOP decreased to a normal level, however, the ANVs remained (Figure 3E). The perpendicular part of ANV did not disappear even after the administration of topical betamethasone was continued for more than a few months or a year (Figure 3C,E). Prior to the start of the betamethasone treatment, the circumferential area of ANV was sometimes hidden by granuloma (Figure 3F,G), or invisible. Among the five eyes that showed negative FGA, two eyes showed tiny small nodules in the angle without KPs, and two cases had only PAS and previous chorioretinal atrophy. The remaining one case was diagnosed as coexistence of primary open-angle glaucoma and general sarcoidosis.

FFA detected abnormalities in 40 (87%) of the 57 eyes, all of which had PVN (80%) (Figure 4) and/or nodules in the retina or choroid (70%) (arrowheads in Figure 2B). The abnormalities observed in panoramic FFA were PVN, nodules in the retina known as ‘candle-wax drippings’ (Figure 5A,B) in the optic papilla (Figure 5C,D) and the choroid (Figure 6A–C), chorioretinitis (stars in Figure 2B), chorioretinal atrophy (Figure 6A,C), and ME (Figure 7A,B). PVNs were mostly observed along the veins, but were also found along the artery. The nodule in the PVN seemed to push the vein wall (Figure 4B), and postcapillary venules (PCVs) adjacent to the PVN showed dilatation (Figure 4C) and leakage of fluorescein dye (Figure 4D). In the sarcoidosis patients, FFA provided much more information than the fundus examination, especially in the extreme peripheral area (Zone III) (Figure 2B). Of the three zones, the abnormalities were most frequently found in the Zone II area. However, five eyes showed abnormalities only in Zone III, and three eyes showed abnormalities only in Zone I. In 2 (9%), one (6%), and three (25%) eyes in Groups I, II, and III, respectively, FFA examination detected no abnormalities.

### 3.1. Complications and Treatments

Although two patients were prescribed oral steroid medication from the Department of Internal Medicine at our institution, that medication was not prescribed for an ophthalmology-related reason. In only one patient, repeated retrobulbar injection of triamcinolone was administered for the treatment of vitreous opacity following tube shunt surgery, however, the VA in that patient remained at less than 1.00 logMAR units for 6-years postoperative. Thirty-nine eyes (69%) showed an elevated IOP of more than 21 mmHg at the initial visit. Of those 39 eyes, 32 were prescribed topical betamethasone, and the IOP in 25 eyes decreased and stayed within normal limit. There were 14 eyes (25%) that underwent glaucoma surgery; seven of those 14 eyes showed an IOP of more than 20 mmHg, despite the topical betamethasone treatment, and the other seven eyes were not prescribed topical betamethasone treatment prior to TRAB due to being at the end stage of visual field deterioration. 

We investigated the risk factors for glaucoma surgery in the clinical manifestation and found that VFD in A–G classification (*p* = 0.0349) and FFA in Zone III (*p* = 0.0484) were the significant risk factors (Table 5). Thirteen eyes showed response to steroids, as evaluated by the method previously described by Armaly [17]. However, one eye showed an elevation of IOP to 35 mmHg from the normal level when the topical betamethasone treatment was changed from 3-times daily for 4 years to 6-times daily due to the anterior chamber inflammation not being sufficiently suppressed via the 3-times-daily treatment. It should be noted that in most of the eyes, the anterior chamber inflammation was sufficiently suppressed and IOP remained normal via the 3-times-daily topical betamethasone treatment. In three eyes, the continuous 3-times-daily topical betamethasone treatment was insufficient for suppressing or avoiding the recurrence of anterior chamber inflammation. One eye underwent combined TRAB with phacoemulsification and intraocular implantation, three eyes underwent the combination of TRAB and Molteno or Baerveldt tube-shunt surgery. The indication for combined TRAB and tube-shunt surgery was an elevated IOP with more than 40 mmHg and/or advanced VFD of more than Stage 5 in the A-G classification [18], in addition to the eyes showing response to steroids. All eyes that underwent the combined surgeries of TRAB and tube-shunt implantation were free from IOP elevation and anterior chamber inflammation with continuous 3-times-daily topical betamethasone administration in the postoperative periods, despite of the fact that two patients were diagnosed as moderate steroid responders. 

ME was found in four (9%) of the 46 eyes in which FFA examination was possible. BCVA in those eyes with ME ranged between 0.1 and 0.2 logMAR units, and no special treatment for ME was performed in those four patients. Thirteen patients underwent retinal photocoagulation due to RVO (four eyes, 7%) and retinal degeneration (one eye, 2%)/tear (eight eyes, 14%). There was one eye with narrowing of the retinal vein in Zone II (arrowheads in Figure 7C) or Zone III. RVO was found in one eye in Zone III, two eyes in Zone II, and two eyes in Zone I (Figure 8A,B). Narrowing of the vein or RVO in sarcoidosis showed a long length of affected vein (between arrowheads in Figure 7C and Figure 8A,B). FFA in the eyes that underwent photocoagulation for retinal degeneration/tear detected abnormalities in the retinal blood vessel revealed a non-perfusion area and NV (arrow and encircled area in Figure 9A), angioma near the retinal tear (arrows and arrowhead in Figure 9B), and chorioretinal atrophy (Figure 9C). Among 27 eyes that underwent cataract surgery, 10 patients underwent the surgery in both eyes during the observation periods and VA was recovered in all eyes post-surgery. 

### 3.2. Pathological Findings of the Eyes

Pathological investigation was performed in five autopsy cases of sarcoidosis and 14 TRAB specimens with secondary glaucoma due to sarcoidosis and a vitrectomy sample from one case of histologically diagnosed sarcoidosis using light, transmission, and scanning electron microscopy. All patients that underwent TRAB were included in the clinical part of this study. The patient age and autopsy cases are listed in Table 1B. In the autopsy eyes, the granuloma was found in the optic nerve. Optic nerve fiber bundles were replaced by fibrotic tissue (stars in Figure 10A) in the area where granuloma surrounding the blood vessel was found (Figure 10B). The choroid became of uneven thickness and was also replaced by fibrotic tissue in the area of chorioretinal atrophy (Figure 11). Granulomas in the retina were often found adjacent to blood vessels (Figure 12A and Figure 13A,B), and were composed of epithelioid cells that had subplasmalemmal linear density (arrowheads in Figure 12B inset). The thick wall of the blood vessel was observed not only in the retinal artery (Figure 12A), but also in the vein (arrowheads in Figure 13A,B), and the thickened walls of those vessels was composed of basal lamina layering (stars in Figure 12C). In the tangential sections of the retina, the relationship between the granulomas and the retinal blood vessels (Figure 13A) became clear (Figure 13B). The walls of the retinal blood vessels often became thick (arrowheads in Figure 13B), and lumen was occluded (Figure 13C). Macrophages and lymphocytes were often found adjacent to the PCVs (Figure 14A,B) where endothelium became discontinued (Figure 14B). Moreover, the endothelium of the blood vessels often had vacuoles along the endothelial junction (Figure 14C, inset). The granulomas in the retina extended into the vitreous cavity (Figure 15A–C) and formed a snowball opacity (Figure 15F) or string of pearls. Proliferation from the retina to the vitreous (Figure 15D), or vitreoretinal adhesion in the degenerated retina (Figure 15E), was sometimes observed in the peripheral retina.

In six (43%) of 14 eyes, granulomas were found in the TRAB specimens. Granulomas were found in the iris in one eye (Figure 16A), yet were mostly found in the outflow routes along the collector channels (CCs) (Figure 16B), the Schlemm’s canal (SC) (Figure 16B,C), the iris root (Figure 16D), and the TM. No inflammatory cells were found in the outflow routes in the first TRAB (Figure 16D) when the eye had been continuously administered topical betamethasone. In the second TRAB combined with tube-shunt surgery, CD68-positive cells and lymphocytes appeared in the SC and CCs after the discontinuation of the betamethasone administration due to steroid-induced IOP elevation (Figure 16F). No inflammation of the anterior chamber or IOP elevation was observed after the second surgery of combined TRAB and tube-shunt implantation, even with continuous topical betamethasone administration. One patient with a non-steroid responder eye discontinued the topical betamethasone administration for 21 months, and then returned to our hospital with an IOP of 38 mmHg and advanced VFD of Stage V in A-G classification. In that eye, PAS of 50% was observed before TRAB combined with tube-shunt surgery, and of all eyes examined, it was the only eye with PAS of more than 25%. Pathological investigation of the TRAB specimens revealed SC occlusion (Figure 16G). No recurrence of anterior chamber inflammation and no IOP elevation was observed post glaucoma surgery with continuous 4-year administration of topical betamethasone. Granuloma in the SC occluded the SC lumen (Figure 16H), and then tended to disappear, leaving the SC occluded (Figure 16I). Although no granuloma was found in the eye of Autopsy Case 3, NV in the TM was observed. In the eyes of Autopsy Case 4, no compatible change with sarcoidosis was observed. 

## 4. Discussion

The findings in this study showed that it was possible to observe nodules on the TM via gonioscopy in most of the eyes (Figure 1), except in six eyes (12%) that showed positive for ANV in FGA yet no visible nodule on the TM (Figure 1C, inset, and Figure 3A,D). In this study, FGA detected ANV in 52 eyes (91%). In regard to the characteristic ocular signs in cases of ocular sarcoidosis, TM nodules/PAS are reported to be the highest sensitivity and the second-highest specificity [7]. Thus, and combined with our FGA findings of the high frequency of ANV in our cases, FGA may be a powerful tool for the diagnosis of sarcoidosis. However, although FGA using a fundus camera was easy to manipulate [14], FGA using a slit lamp required a special technique and instrument. The frequent findings of bleeding in the TM was an important sign of ANV (Figure 2A, inset), and TM staining with bleeding was clearly differentiated from blood reflux in the SC when the gonioscopy pressure was released. The configuration of ANV in sarcoidosis was similar with that in ischemic retinal disease, except is some cases of sarcoidosis-related ANV (Figure 3G). In the former case, FFA can easily differentiate the area of the retina with no ischemic sign in the posterior part (Figure 1B). One possible reason for positive ANV but no visible nodule may be due to the nodule being located deeply distant from the surface of the TM, such as in CCs (Figure 16B), in the SC (Figure 16B,C,H,I), or in the TM [9]. Five cases (9%) showed negative with FGA. This may be due to the initial stage of the nodules or an angle with very old inflammation. In a previous pathological study, we reported ANV in the eyes of patients with sarcoidosis [9]. In a previous study by Meyer et al. [19], the authors reported that macrophages obtained from bronchoalveolar lavage fluid in patients with sarcoidosis induced markedly greater degree of angiogenesis than those from normal healthy volunteers [19]. It is obvious that granuloma induces ANV from the iris root, and that ANV extends toward granulomas. Although it may be of interest that all eyes with KPs on the cornea can have nodules on the TM, there were no eyes in this present study with KPs on the cornea without nodules on the TM. Published literature has described “Occult KP”, a term indicating cases in which there are nodules in the TM, yet no inflammation in the anterior chamber [20]. Thus, a question arises as to where the macrophages came from. Our findings suggest that macrophages in the TM or anterior chamber may be supplied from the SC and CCs (Figure 16B,F) via the aqueous vein. Macrophage infiltration into the CCs or the SC strongly supports this notion, and inflammation in the SC may be better termed as “Schlemm canalitis” [9]. This notion may be also applied to another type of granulomatous inflammation in Posner-Schlossman syndrome due to the extremely high IOP before inflammation in the anterior chamber becoming visible.

FFA appears to be an excellent method for detecting abnormalities in ocular sarcoidosis, as it was able to detect PVN and/or nodules in 40 eyes (87%) in this study. In addition, all eyes that showed positive manifestations in FFA had PVN (80% as a whole) (Figure 4) and/or nodules (arrowheads in Figure 2B and Figure 5A,B) in the retina or choroid (70% as a whole). PVN and nodules can be termed as granuloma-associated manifestations in FFA. The granulomas (nodules) in the posterior part of the eyes caused a variety of secondary complications. Granuloma in the optic nerve (Figure 10) caused a VFD similar to that in glaucoma, however, there was no glaucomatous cupping at the initial stage. The reason for the VFD similarity may be due to the fact that the optic-nerve fiber bundles were segmentally damaged (Figure 10). From the FFA results, it is speculated that the ’candle-wax drippings’ were superficial retinal exudate, as there was no capillary pattern [21]. However, this seemed more likely to be the nodule itself (arrowheads in Figure 5A,B and Figure 7A), as no exudative change was observed in the pathological examination of the candle-wax dripping (nodule/granuloma) in the autopsy eyes (Figure 12A, Figure 13A,B and Figure 15A). PVN associated with the retinal vessel (Figure 4B) also appeared similar to candle-wax dripping. Thus, the question is whether or not positive/negative fluorescein leakage may depend on the stage of granuloma formation and the location of the granuloma. When the granuloma arose from the choroid, it tended to become much bigger than those in the retina (Figure 6), and there may be no choroidal filling in the early stage in IGA [11,22], as leakage showed by FFA in the late stage (Figure 6A–C). The wide range of chorioretinal atrophy composing the thick and thin fibrotic tissue observed in the pathological examination (Figure 11) may be compatible changes with those in Figure 6. The retinal granuloma (nodule) showed ellipsoid or round-shaped staining in the late stage by FFA (Figure 4A, Figure 5B and Figure 7A). The primary reason for the leakage in the granuloma may be because PCVs were involved (Figure 4C,D) and that the endothelium of the PCV had separation of the junction where inflammatory cells were invaded (Figure 14A,B). It should be noted that the new technology enabled for a wider-field fundus view to be obtained [23] without mounting photographs. However, even via the use of a new fundus camera, it is still impossible to obtain detailed information about the dilatation and fluorescein leakage of PCV, as was observed in our study (Figure 4C,D). The endothelium of retinal blood vessels had vacuoles along the junctional areas (Figure 14C), which may be another reason for the fluorescein leakage in the non-granulomatous area. ME may also be caused by endothelial damage in retinal micro blood vessels that have no smooth muscle (PCV in Figure 14A,C). Although PVNs were observed mostly in veins, they were also sometimes observed in the retinal arteries (Figure 12A). Moreover, the retinal veins were more susceptible to occlusion (Figure 8 and Figure 13C). The clear difference between RVO caused by arterial sclerosis and RVO in sarcoidosis is that the former occurs at the crossing part of the retinal artery. On the other hand, the narrowing (Figure 7C) or occluded area (Figure 8A,B) in the eyes with sarcoidosis was much longer. There may be two reasons for such a long area of vascular narrowing or occlusion in ocular sarcoidosis. Although some PVNs disappeared without leaving any abnormality despite no systemic steroid medication (Figure 4A, inset), granuloma in the PVN sometimes caused narrowing of those vessels (Figure 13A), and extreme thickening of the vessel wall (Figure 12A, arrowheads in Figure 13A–C) due to basal lamina layering (Figure 12C) [13] over a certain length of those blood vessels (Figure 13B). Sheathing of the arterial alteration [12] in patients with sarcoidosis may also be caused by the basal lamina layering. The components of the ’string of pearls’ or ’snowball’ opacity in the vitreous were the same as that of granuloma (Figure 15F), and those may have been supplied from granuloma or macrophages in the retina (Figure 15A–C).

### 4.1. Complications and Treatments

ME, RVO, a retinal tear, and glaucoma are the primary serious complications in cases of ocular sarcoidosis. In the published reports from different countries, the frequency of ME in sarcoidosis cases varied, and ranged from 9% [2], to 28% [24], to 44% [3], and 58% [1]. In this present study, FFA examination detected ME in only 9% of the cases. It should be noted that racial differences may have been one possible reason for those varied results. It should be noted that the frequency of retinal vasculitis in sarcoidosis has previously been reported as vasculitis (37%) [2], periphlebitis (29%) [25], and nodular periphlebitis (45%) [7], and that the reports of RVO are rare [26]. In the present study, four cases (7%) had undergone retinal photocoagulation due to RVO. FFA could detect RVO more easily when compared to ordinal fundus examination, and was good for being used as a compass to judge the ischemic area at the time of laser photocoagulation (Figure 8A,B).

Vitreoretinal proliferation (Figure 15D), or adhesion (Figure 15E), were often observed in the autopsy eyes, and were composed of fibroblast-like cells extending from the retina to the vitreous (Figure 15D). It has previously been reported that retinal breaks develop in 3.7% of patients with symptomatic isolated posterior vitreous detachment [27]. In our patients, a retinal tear (Figure 9B,C) was found in eight eyes (14%) and was treated by laser photocoagulation. This finding indicated that the frequency of retinal tears in sarcoidosis is much higher than that in the normal healthy population. Possible reasons for the high risk of a retinal tear in sarcoidosis cases may be the proliferation by inflammatory cells into the vitreous (Figure 15D), the vitreoretinal adhesion in the degenerated retina (Figure 15E), and the abnormal vascular changes (Figure 9A–C). In this study, the number of eyes with low-vision BCVA was much less than that in previous reports [4,24]. In fact, only three eyes showed a BCVA of more than 0.4 logMAR units (two eyes with advanced-stage glaucoma, and 1 eye with a small pupil with posterior synechia and cataract). The reason for this may be that in comparison with the other studies, there were no eyes with serious ME and intense vitreous opacity [4,22,28]. There were two eyes that had undergone repeated retrobulbar injection of triamcinolone due to vitreous opacity. Vitrectomy had been performed in one eye of a patient (biopsy Case 6) who visited after the FGA and FFA examination period. In another patient, repeated retrobulbar triamcinolone injections were performed, yet vision remained at less than 0.00log MAR units.

In this study, although the superficial optic-disc nodules were not difficult to find (Figure 5C,D) [29], granuloma in the optic nerve (Figure 10A) was impossible to find, as VFDs in the optic nerve seemed to be similar with those in glaucoma. However, VFD due to granuloma in the optic nerve may appear in advance of the glaucomatous cupping becoming clear. Twenty-one eyes (37%) were diagnosed as secondary glaucoma due to sarcoidosis, and there were 20 eyes with open-angle glaucoma with less than 25% PAS and 1 eye with 50% PAS (Figure 16G). In the previous studies [1,2,3,4,5], the ratio of the occurrence in secondary glaucoma due to sarcoidosis reportedly ranged from 5.7% [5] to 35.29% [3], lower than in this present study. Our findings showed that glaucoma surgery had a significant correlation with VFD in A–G classification (*p* = 0.0349) and the number of FFA abnormalities in Zone III (*p* = 0.0484) (Table 5). Those results indicate that the patients who needed glaucoma surgery had a serious VFD and more abnormalities in Zone III. Although it is thought that the vascular system in the anterior segment, including the SC and aqueous veins, is dominated by a different vascular system from that in Zone III, there may an interconnecting vascular system. In the 21 eyes with VFD, TRAB and TRAB combined with tube-shunt surgery was performed in 11 and 3 patients, respectively. TRAB was useful for the diagnosis of sarcoidosis, as granulomas were found in the TRAB specimens obtained from six (43%) of 14 eyes. All eyes that received TRAB combined with tube-shunt surgery were free from IOP elevation following continuous 3-times-daily topical betamethasone administration, despite of the fact that two eyes of two of those patients were steroid responders. Thus, tube-shunt surgery may have the power of escaping steroid response [30]. Due to the recurrent nature of ocular sarcoidosis, continuous topical betamethasone administration is often required. In our study, only two patients underwent oral steroid administration, which was much fewer compared to the other studies [4,31]. Moreover, our findings revealed that maximum IOP had a significant correlation with nodules in the angle (*p* = 0.02696, Table 4) and VFD (*p* = 0.0151, Table 5). Based on those results, it is obvious that nodules in the angle should be treated to avoid the progression of VFD. However, the reason for maximum IOP not being a risk factor for glaucoma surgery (*p* = 0.1330, Table 5 may be due to the fact that we principally prescribed topical betamethasone to the eyes with high IOP, thus resulting in decreased IOP. 

Although tent-like PAS is known to be a typical change in ocular sarcoidosis, that change may not result in IOP elevation. On the other hand, wider PAS, which we experienced in the right eye of Case 27, did seem to be one of the reasons for elevated IOP (Figure 16G). Wide-range PAS or the proliferation of melanocytes on the surface of the TM may cause insufficient aqueous outflow, thus resulting in occlusion of the SC. However, normal IOP is often experienced in primary angle-closure glaucoma eyes in which PAS reaches 50%. Grant [32] described in his textbook that IOP still remains normal in eyes with a PAS of 50% if the aqueous outflow facility in those eyes is “good normal”. The eyes of Case 27 may have had “good normal” aqueous outflow facility, as the IOP in that case had remained at 12–14 mmHg for 18 years (see the legend in Figure 16G) before the onset of IOP elevation. Therefore, the high IOP of more than 30 mmHg after the occurrence of PAS could not be explained only by 50% PAS. The gradual increase in IOP before the occurrence of wide PAS suggested that SC occlusion may be a primary reason for the uncontrolled IOP in the right eye of Case 27 (Figure 16G–I). Since the SC originates from the blood vessel in the embryonic stage [33], SC occlusion (Figure 16G–I) caused by inflammation (Figure 16B,F) or granuloma (Figure 16C,H,I) may be a part of the microangiopathy in ocular sarcoidosis. Therefore, the most important reason for continuous topical betamethasone administration may be to suppress IOP elevation at the time of recurrence, as well as to avoid the irreversible changes of SC occlusion (Figure 16G–I) [9] in addition to PAS and the proliferation of melanocytes on the surface of the TM (Figure 16G).

### 4.2. Study Limitations

This current study did have some limitations. For example, it should be noted that there might have been selection bias regarding the patients used in this study, as those patients were recruited from a healthcare center or a tertiary referral from an ophthalmology clinic. Moreover, it was not possible to wholly determine the frequency of the various types of intraocular manifestations from the sarcoid patients in this study. In addition, the number of cases in which it was possible to examine both FGA and FFA was limited. Thus, these factors may have caused selection bias. 

## 5. Conclusions

The concept of microangiopathy in ocular sarcoidosis is the “NV” in ANV, without a non-perfusion area in the posterior part of the eye, as was frequently observed in FGA, as well as the “endothelial damage” in the SC, CCs, and micro blood vessels lacking of smooth muscle in the posterior part of the eye, as was observed via FFA and the pathological investigation of this study. Microangiopathy in ocular sarcoidosis may cause serious complications, such as retinal tears, RVO, ME, and secondary open-angle glaucoma. Our findings show that FGA and FFA are powerful tools for the diagnosis and proper treatment of ocular-sarcoidosis-related complications, and that continuous topical betamethasone or corticosteroid administration may be important for avoiding irreversible changes of SC occlusion and wide range of PAS. 

## Figures and Tables

**Figure 1 diagnostics-11-00039-f001:**
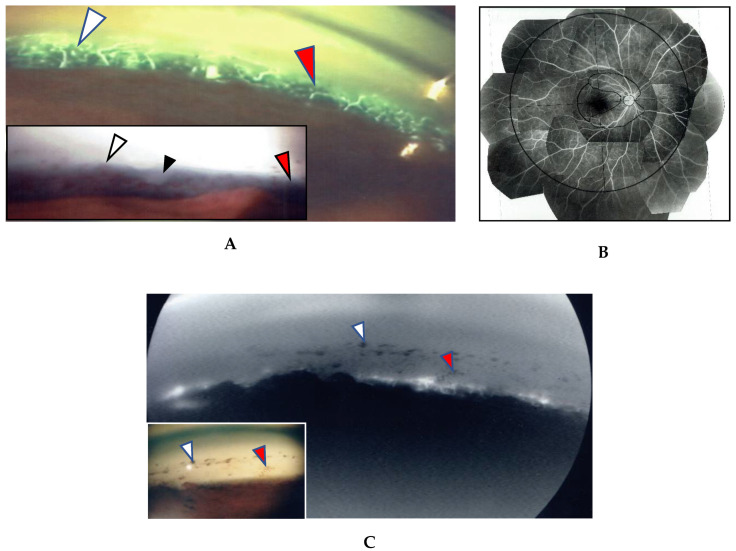
(**A,B**) Images of the right eye of Case 53, a 50-year-old female in Group III with a max intraocular pressure (IOP) of 14 mmHg. Photographs of fluorescein gonio angiography (FGA) and fundus fluorescein angiography (FFA) mounted on a transparent sheet showing different zones (**B**) and slit-lamp gonioscopy (**A**, inset). A nodule can be seen at the ciliary body band (black arrowhead in **A**, inset). No topical betamethasone was prescribed prior to FGA and gonioscopy. The corresponding areas in **A** and **A**, inset are indicated by white and red arrowheads. Angle NVs can be seen arising perpendicularly from the iris root, and then changing their direction and running circumferentially along the trabecular meshwork; (**B**) Panoramic FFA image showing no non-perfusion area. The patient has had no recurrence of inflammation in both eyes for 18 years; (**C**) Fundus-camera images of FGA in the left eye of Case 23, a 72-year-old male in Group I with a max IOP of 35 mmHg. (**C**, inset) Slit-lamp image showing no nodule. Topical betamethasone had been irregularly prescribed. The corresponding areas in **C** and **C**, inset are indicated by white and red arrowheads.

**Figure 2 diagnostics-11-00039-f002:**
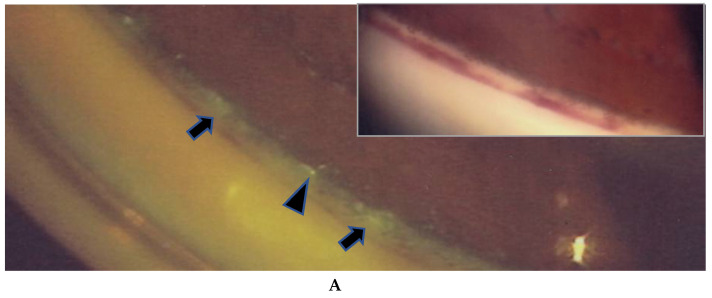
(**A**,**B**) Images of the right eye of Case 46, a 73-year-old female in Group III with a max IOP of 47 mmHg. Photographs of FGA (**A**) and panoramic FFA (**B**) mounted on a transparent sheet indicating different zones. (**A**, inset) Slit-lamp gonioscopy image showing bleeding, possibly from angle NV at the same area as that of FGA. Angle NV observed by FGA showing dot fluorescein staining (arrowhead in **A**) and leakage (arrows in **A**). Topical betamethasone was not prescribed prior to FGA; (**B**) Panoramic FFA image showing perivasculitis (arrows), granulomas (arrowheads), and chorioretinitis (stars) mostly located in Zone III (extreme peripheral area).

**Figure 3 diagnostics-11-00039-f003:**
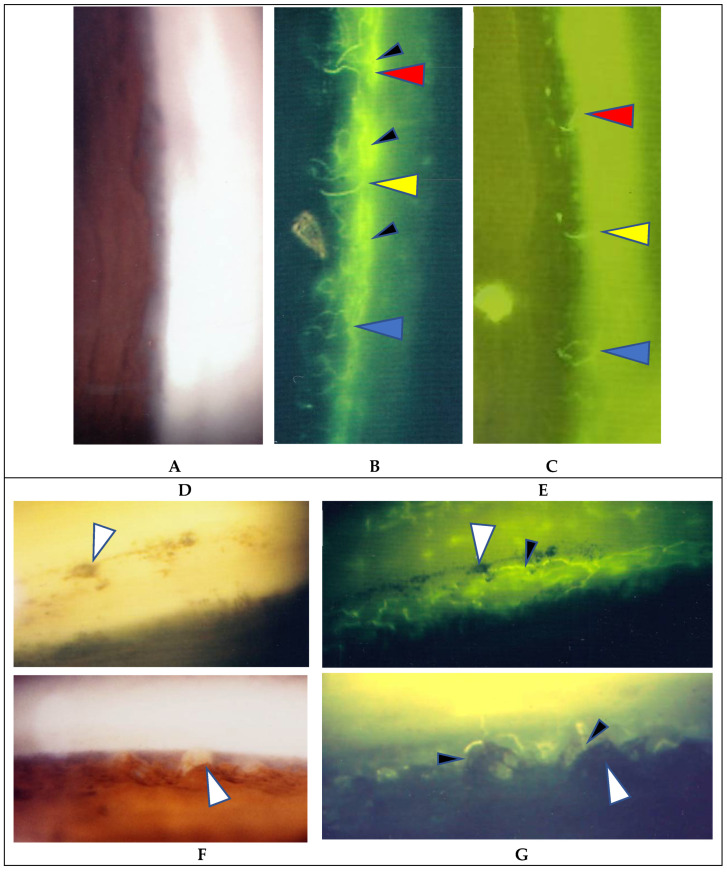
(**A**–**C**) Images of the right eye of Case 54, an 18-year-old male in Group III with a max IOP of 16 mmHg; (**D**,**E**) Images of the left eye of Case 4, a 52-year-old male in Group 1 with a max IOP of 31 mmHg; (**F**,**G**) Images of the right eye of Case 3, and 22-year-old male in Group 1 with a max IOP of 18 mmHg. Gonioscopy photographs taken (**A**) before and (**D**) after the start of topical betamethasone. Granuloma was not found prior the medication being applied (**A**). In Case 54 (**C**) and Case 4 (**E**), angle NV (ANV) remained at 0.8 and 1.6 years, respectively, after the start of a 3- to 4-times daily continuous topical betamethasone administration. In Case 4, IOP lowered from 31 to 14 mmHg (**E**) and the nodules disappeared (**D**) after the start of topical betamethasone, yet FGA clearly demonstrated ANV (**E**). Changes observed after the start of the topical betamethasone in both eyes were that the circumferential parts of ANV (small black arrow heads in B) almost disappeared (**C**) or showed no leakage of fluorescein die (small arrowheads in **E**). Red, yellow, and blue arrowheads in **B** and **C** indicate the respective corresponding areas. In other cases, the circumferential areas of ANV were not visible due to being blocked by nodules (small arrowheads in **F**) or being obscured by the trabecular meshwork. Large white arrowheads in **D** and **E**, and **F** and **G** indicate corresponding areas, respectively. No topical steroid was prescribed at the time of FGA (**G**).

**Figure 4 diagnostics-11-00039-f004:**
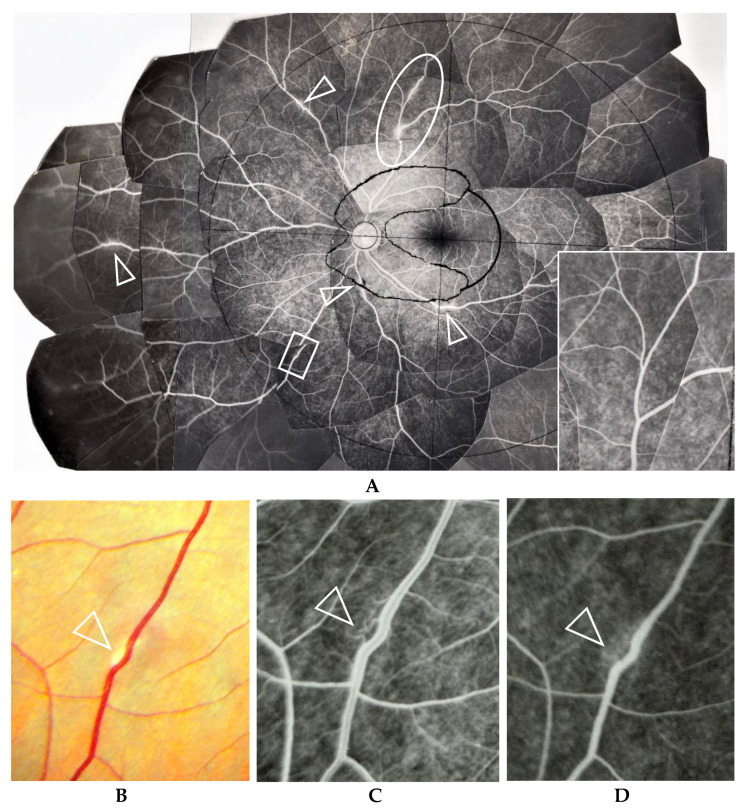
(**A**–**D**) Images of the right eye of Case 40, a 23-year-old male in Group II with a max IOP of 14 mmHg. Photographs of FFA mounted on a transparent sheet with different zones (**A**) and fundus (**B**). Photograph **C** and **D** (early stage (**C**) and late stage (**D**) of fluorescein injection) are high magnification images of the boxed area in (**A**). The FFA photograph in **A**, inset is the oval-shape encircled area shown in **A** taken 2.8 years after the FFA of A. Nodules in the retina completely disappeared in 2.8 years (**A**, inset) with no systemic steroid medication. Arrowheads in **A**: perivascular nodules. Postcapillary venules became dilated (arrowhead in **C**) and showed leakage (arrowhead in **D**) in the nodule adjacent to the vein (arrowhead in **B**).

**Figure 5 diagnostics-11-00039-f005:**
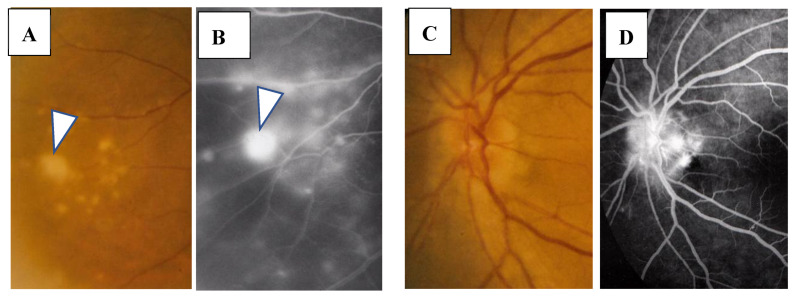
(**A**–**D**) Images of the right eye of Case 32, a 55-year-old female in Group II with a max IOP of 18 mmHg. Photographs of the fundus (**A**,**C**) and FFA (**B**,**D**). Nodule (arrowhead in **A**) shows almost the same size of fluorescein leakage (arrowhead in **B**). Nodules in the optic nerve head (**C**) show a granular staining pattern of fluorescein leakage (**D**).

**Figure 6 diagnostics-11-00039-f006:**
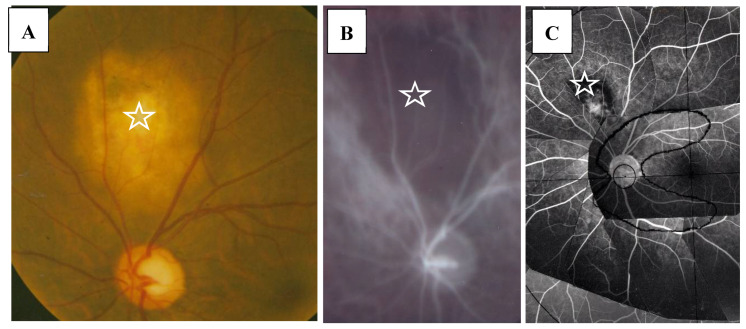
(**A**–**C**) Images of the right eye of Case 9, a 33-year-old male in Group I with a max IOP of 12 mmHg. Photographs of the fundus (**A**), indocyanine green angiography (**B**), and FFA on a mounted transparent sheet with different zones (**C**). Giant nodule located in the choroid (stars in **A**–**C**) showing no filling of indocyanine green (**B**). Fluorescein leakage can be seen at the center of the giant nodule (**C**).

**Figure 7 diagnostics-11-00039-f007:**
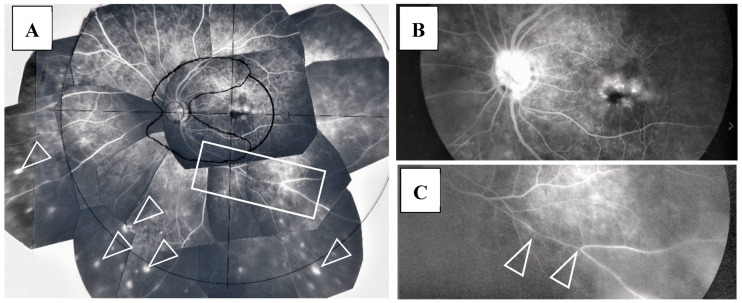
(**A**–**C**) Images of the left eye of Case 5, a 64-year-old female in Group I with a max IOP of 28 mmHg. Photographs of FFA mounted on a transparent sheet with different zones (**A**) and FFA in the late stage (**B**,**C**). Abnormalities of the retina existed from Zone I to III. Numerous nodules (arrowheads in **A**) were found mainly in Zone III. Diffuse fluorescein leakage of the optic nerve head and macular edema were found in the late stage (**B**). Image **C** is an enlargement of the boxed area in **A**. The retinal vein became very narrow with a long length (arrowheads in **C**).

**Figure 8 diagnostics-11-00039-f008:**
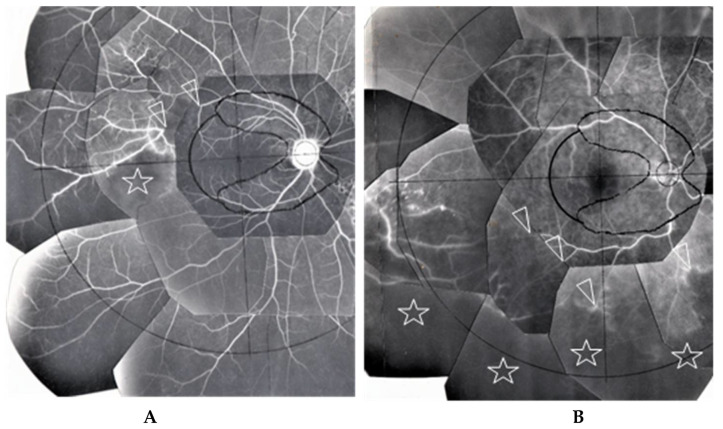
(**A**) Image of the right eye of Case 51, a 31-year-old male in Group III with a max IOP of 22 mmHg; (**B**) Image of the right eye of Case 6, a 65-year-old male in Group I with a max IOP of 28 mmHg. The FFA photographs were mounted on a transparent sheet with different zones. Flow in the retinal vein was abruptly stopped for 2–3 optic disc diameters (between the two arrowheads in **A**). The avascular area in **B** (stars) was wider than that in **A** (star) because of 4 branch veins (arrowheads in **B**) that were occluded. Retinal photocoagulation was performed in both eyes of the patients for the nonperfusion areas. Vascular occlusion (stars in **A** and **B**) occurred mainly in Zone II and III (**A**,**B**), respectively. The right eye of Case 6 had undergone trabeculectomy.

**Figure 9 diagnostics-11-00039-f009:**
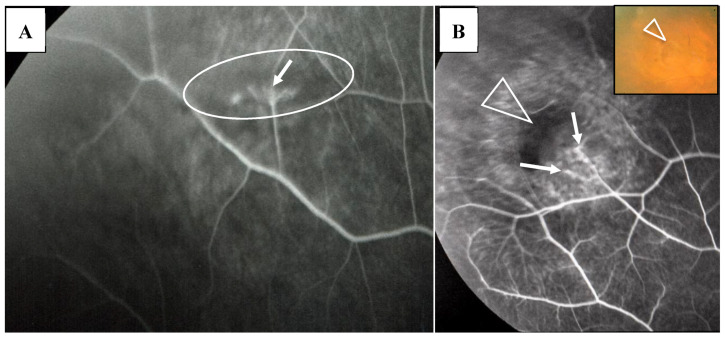
(**A**) Image of the right eye of Case 3, a 22-year-old male in Group I with a max IOP of 18 mmHg; (**B**) Image of the right eye of Case 12, a 22-year-old male in Group I with a max IOP of 15 mmHg; (**C**) Image of the right eye of Case 29, a 77-year-old male in Group II with a max IOP of 35mmHg. Photographs of FFA (**A**,**B**) and FFA mounted on a transparent sheet with different zones (**C**). Neovascularization (arrow in **A**) and occlusion of the blood vessels (arrows in **B**) were observed in the lattice-like degeneration (encircled area in **A**) and adjacent to the retinal tear (arrowhead in **B** and **B**, inset), respectively. There was local retinal detachment with fluorescein staining of pigment epithelium (**B**). Retinal photocoagulation was performed for the retinal tear (arrowhead in **C**) in the chorioretinal atrophy (star in **C**).

**Figure 10 diagnostics-11-00039-f010:**
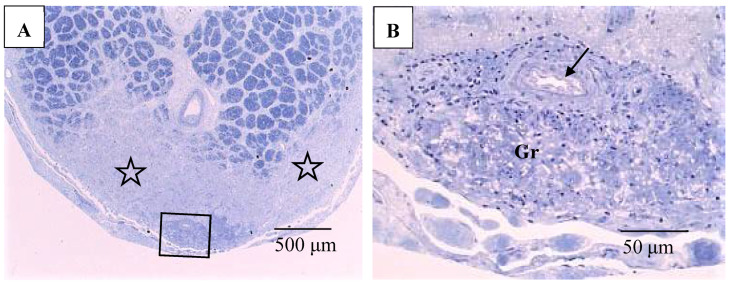
(**A**,**B**) Light microscopy photographs of the optic nerve in the right eye of Autopsy Case 1, a 75-year-old female with a max IOP of 26 mmHg (Epon embedding, toluidine blue stain). Image B is a higher magnification of the boxed area in **A**. The location of the optic nerve was 2 mm from the optic nerve head. The nerve fiber bundles were replaced by fibrotic tissue (stars in **A**). Granuloma (Gr in **B**) was found close to the pia mater (boxed area in **A**). Infiltration of lymphocytes were observed adjacent to blood vessel (arrow in **B**).

**Figure 11 diagnostics-11-00039-f011:**
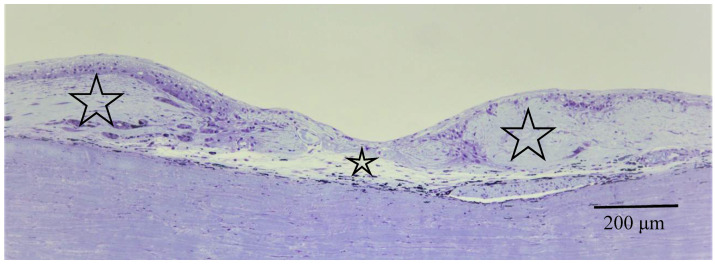
Light microscopy photographs of the peripheral area of the retina and choroid in the right eye of Autopsy Case 2, a 78-year-old female with a max IOP of 22 mmHg (methylmethacrylate embedding, toluidine blue stain). The retina and choroid became very thin (**small star**) at the center of chorioretinal atrophy, which was surrounded by abnormally thick fibrotic tissue in the choroid (**large stars**).

**Figure 12 diagnostics-11-00039-f012:**
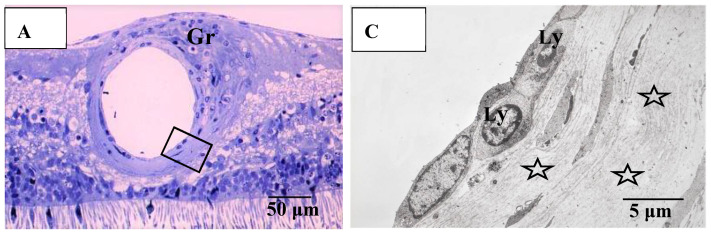
(**A**–**C**) Light microscopy photograph of granuloma (Gr) adjacent to the retinal artery (**A**) located in the area of radial peripapillary capillaries, and transmission electron microscopy photographs (**B**,**C**) in the left eye of Autopsy Case 1, 75-year-old female. **B** and **C** are ultrathin sections of granuloma (Gr) and boxed area in (**A**), respectively. Epon embedding, toluidine blue stain. Granulomas were composed of macrophages (MP in **B**) and had subplasmalemmal linear density (arrowheads in **B**, inset). The inset in (**B**) is a high magnification of the boxed area in **B**. The arterial wall became thick because of basal lamina layering (stars in **C**). Lymphocytes (Ly in **C**) were observed under the endothelium of the artery.

**Figure 13 diagnostics-11-00039-f013:**
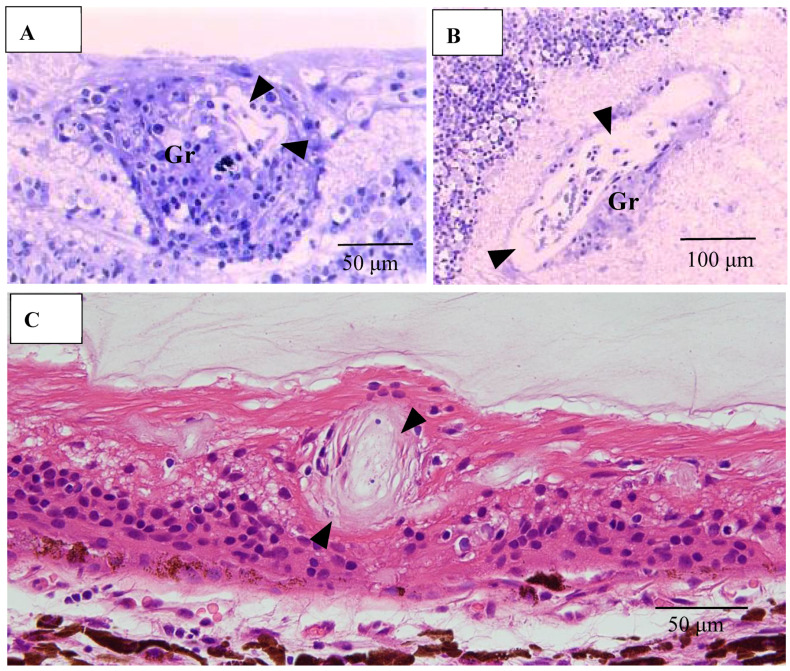
(**A**–**C**) Light microscopy photographs of retinal vein in ordinal (**A**,**C**) and tangential sections (**B**) of the left eye of Autopsy Case 1, a 75-year-old female. (**A**,**B**) Methylmethacrylate embedding, toluidine blue stain. (**C**) Paraffin embedding, PAM stain. Granuloma (Gr in **A** and **B**) accompanying lymphocytes seemed to push the vein wall. The basement membrane seemed to be very thick (arrowheads in **A**–**C**). The blood vessel in the retina was occluded (**C**).

**Figure 14 diagnostics-11-00039-f014:**
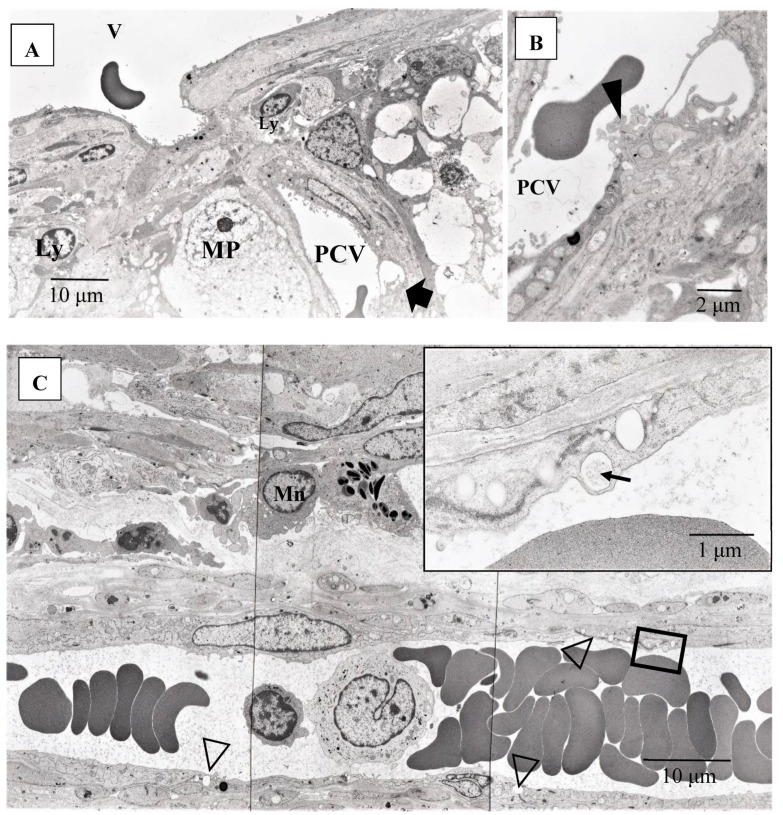
(**A**–**C**) Transmission electron microscopy photographs of retinal blood vessels in tangential sections (**A**–**C**) of the left eye of Autopsy Case 1, a 75-year-old female. Samples of the retina were taken from the area of the radial peripapillary capillaries (**A**,**B**; large magnification of the area pointed by arrow in **A**) and the periphery (**C**). Macrophage (MP) surrounding the postcapillary venule (PCV) and endothelium of the PCV was infiltrated by inflammatory cells (arrowhead in **B**). The endothelium of the retinal blood vessel had many vacuoles (inset in **C**) along the junctional areas (arrowheads in **C**). Some of the vacuoles contained fibrillar materials (arrow in **C**, inset). Ly: lymphocytes, Mn: monocytes, V: retinal vein.

**Figure 15 diagnostics-11-00039-f015:**
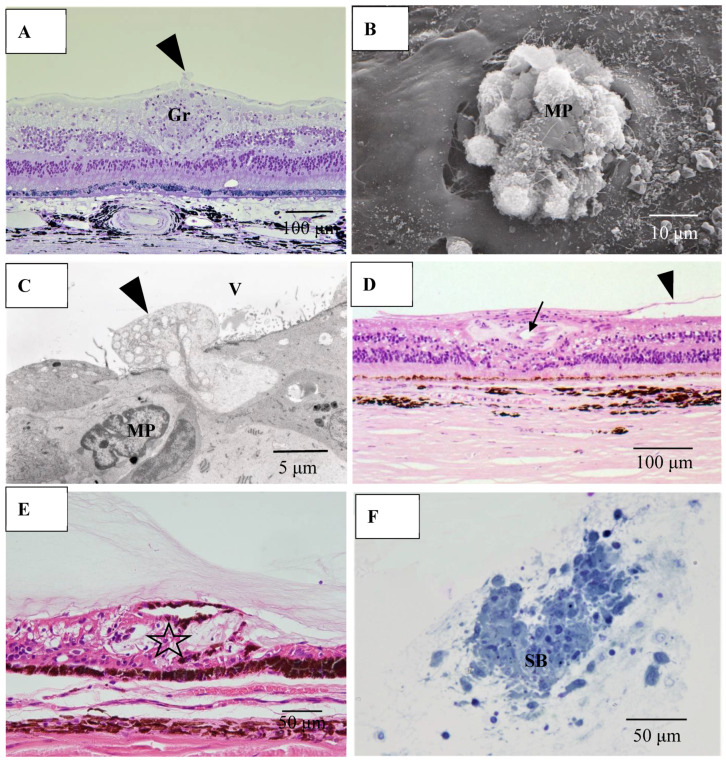
(**A**) Light microscopy photograph granuloma (Gr) in the retina taken from the transitional area from radial peripapillary capillaries to the mid periphery in the right eye of Autopsy Case 2, a 78-year-old female (methylmethacryrate embedding, toluidine blue stain); (**B**) Scanning electron microscopy photographs of granuloma extending from the retina to the vitreous in the right eye of Autopsy Case 1, a 75-year-old female; (**C**) Transmission electron microscopy of granuloma in the left eye of Autopsy Case 1; (**D**,**E**) Light microscopy photographs of the retina in the mid periphery (**D**) and extremely periphery (**E**) in the left eye of Autopsy Case 1. Hematoxylin & eosin stain, paraffin embedding; (**F**) Light microscopy photograph of snowball vitreous opacity obtained from a vitrectomy sample. Epon embedding, toluidine blue stain. Macrophages (arrowheads in **A**, **C** and MP in **B**, **C**) invading from retina into the vitreous (V). Proliferative tissue (arrowheads in **D**) consisting of fibrotic tissue with lymphocytic infiltration adjacent to the retinal vein (arrow in **D**) extending from the retina to the vitreous. Vitreoretinal adhesion where the retina became degenerated (star in **E**) was observed in the extreme peripheral area. The snowball (SB in **F**) opacity is granuloma. FFA showed diffuse and strong leakage from blood vessels (figure not shown).

**Figure 16 diagnostics-11-00039-f016:**
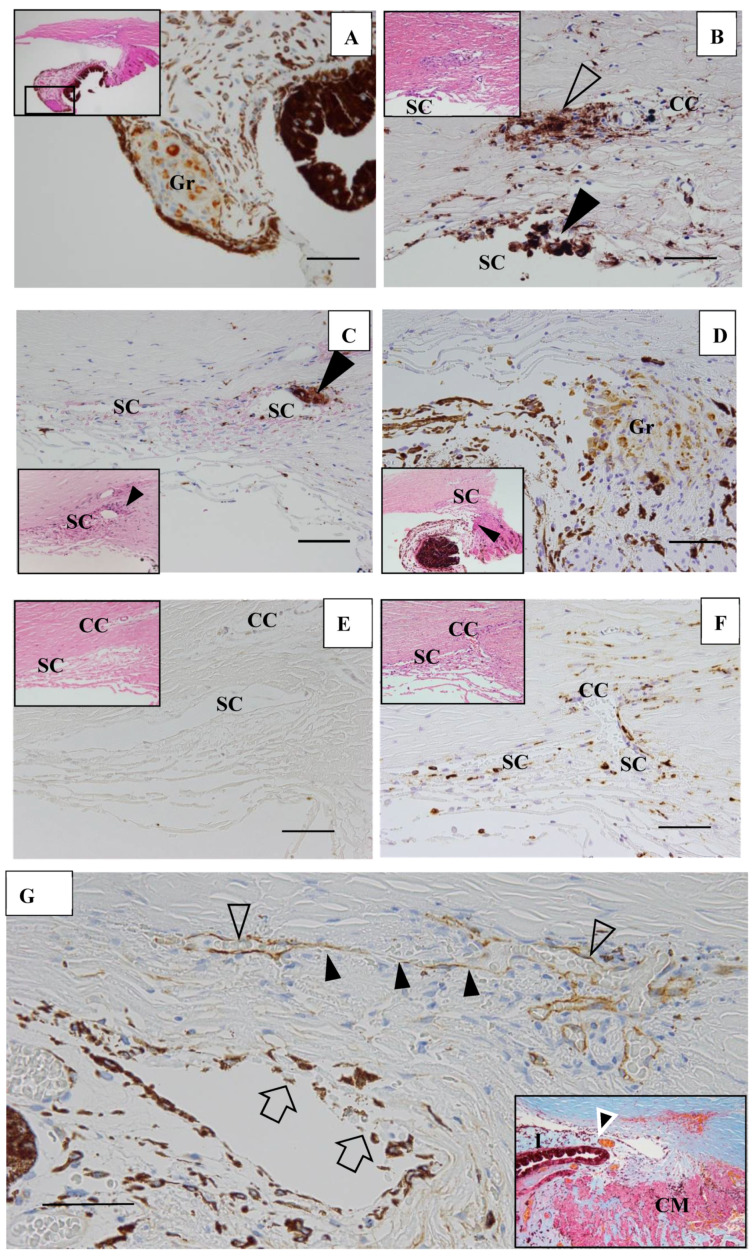
(**A**–**F**). Light microscopy photographs of the angle in trabeculectomy (TRAB) specimens. (**A**–**F**) immunohistochemical staining of CD68. (**A**–**F** inset) Hematoxylin eosin stain of paraffin embedding. (**A**) Image of the right eye in Case 24, a 39-year-old female in Group I with a max IOP of 34 mmHg; (**B**) Image of the right eye in Case 21, a 72-year-old female in Group I with a max IOP of 32 mmHg; (**C**) Image of the left eye in Case 23, a 72-year-old male in Group I with a max IOP of 35 mmHg; (**D**) Image of the left eye in Case 20, a 72-year-old male in Group I with a max IOP of 50 mmHg; (**E**,**F**) Image of the left eye in Case 44, a 74-year-old male in Group III with a max IOP of 54 mmHg. Scale bar: 50 μm. Granuloma (Gr) was found in the pupil area, around the collector channel (CC, open arrowhead in **B**), in the Schlemm’s canal (SC, solid arrow-heads in **B** and **C**), and the iris root (Gr in **D**, arrowhead in **D** inset). CD68-positive cells were not found with a continuous topical betamethasone (CTB) in the first TRAB (**E**), but were found around the SC, collector channel (CC), and in the trabecular meshwork in the second TRAB. This eye was a dilemma case of super mild steroid responder. When CTB (4 to 6 times daily) was continued for 7 months, IOP temporally decreased from 36 mmHg to 15 mmHg. However, it started to increase to 23 mmHg, so CTB was discontinued for 1 week and IOP increased again to 34 mmHg. The second TRAB was then performed in 4 days after the restart of the topical betamethasone. CD-positive cells and lymphocytes infiltration still remained in the SC and the CCs; (**G**) Light microscopy photographs of the angle obtained from TRAB in the right eye of Case 27, a 33-year-old female in Group II with a max IOP of 38 mmHg (thrombomodulin immunohistochemical stain). The inset is a lower magnification image of photo **G** (Masson trichrome stain). CM: ciliary muscle, I: iris, arrowhead in inset: peripheral anterior synechia (PAS). Scale bar: 50 μm. The patient was referred to our hospital in December 1991 due to BHL. Perivascular nodules in the retina and slight inflammation in the anterior chamber were found. IOP had remained at 12–14 mmHg in both eyes for 18 years. IOP in her right eye started to increase to 20 mmHg from March 2009, and topical betamethasone was then started. However, the patient administered the medication irregularly. PAS with 50% (arrowhead in **G**, inset) occurred at 21 months of absence from our hospital. IOP in her right eye increased to 38 mmHg and VFD worsened from normal (March 2011) to Stage V (January 2016) in A-G classification. Because of no response to IOP (i.e., IOP fluctuated around 30 mmHg) by topical betamethasone and glaucoma medication, TRAB combined with tube-shunt surgery was performed in October 2016. Only a small part of Schlemm’s canal (SC) remained open (open arrowheads) and the center part of SC became occluded (solid arrowheads). There was infiltration of melanocytes on the surface of trabecular meshwork (arrows in **G**). No inflammation in the anterior chamber and no elevation of IOP was observed with continuous topical betamethasone administered 3-times daily for 4-years postoperative; (**H**,**I**) Light microscopy photographs of the angle at the 9-o’clock position in the left eye of Autopsy Case 1 (**H**), a 75-year-old female, and at the 9-o’clock position in the left eye of Autopsy Case 2 (I), a 78-year-old female. The inset in **I** is a high magnification image of the granuloma (arrowhead in **I**). Methylmethacrylate embedding, toluidine blue stain. Granulomas (Gr in **H**, arrowhead in **I**) were found in the Schlemm’s canal (SC). The SC is almost occluded in photos **H** and **I**. The granuloma composed of epithelioid cells appeared to be shrinking and disappearing (**I** inset). Scale bar in **F**, **G**: 50 μm.

**Table 1 diagnostics-11-00039-t001:** (**A**) Patients in the clinical and pathological studies (*n* = 57 patients); (**B**) Patients in the pathological study.

**(A)**			
**Age**	**Gender**	**Mean Observation Period (Years)**	**BCVA**
52.6 ± 19.03	female (54%)	8.78 ± 7.52	−0.34 ± 0.74
BCVA: best-corrected visual acuity in logMAR.			
**(B)**			
**Type of Specimens**	**Age**	**Gender**	**Max IOP (R/L, mmHg)**
TRAB (14 eyes) *1	61.29 ± 17.14	M: 7, F: 7	42.64 ± 10.43 *2
Autopsy Case 1	75	F	26/24
Autopsy Case 2	78	F	22/23
Autopsy Case 3	44	F	14/10
Autopsy Case 4	90	M	/
Autopsy Case 5	67	F	/
Biopsy Case 6 *3	32	F	38 *2

*1: All TRAB eyes belonged to the eyes used in clinical study. TRAB specimens were obtained from 14 eyes of 14 patients. One TRAB-Group eye twice underwent TRAB; *2: IOP data were taken from the eyes of TRAB or vitrectomy; *3: Case 6: biopsy of vitrectomy sample in patients with histologically diagnosed sarcoidosis. Abbreviations: IOP: intraocular pressure; TRAB: trabeculectomy; F: female; M: male.

**Table 2 diagnostics-11-00039-t002:** Patient age, gender, and observation periods in each Group of the clinical study (I: biopsy proven, II: bilateral hilar lymphadenopathy (BHL) positive, III: BHL negative).

Group (*n*)	Mean Patient Age (Years)	Gender	Mean Observation Period (Years)
Group I (24)	53.92 ± 18.88	F = 13, M = 11	9.21 ± 6.23
Group II (19)	48.47 ± 16.88	F = 11, M = 8	8.91 ± 9.13
Group III (14)	55.93 ± 22.23	F = 7, M = 7	7.86 ± 7.62
*p*-value	0.496 *1	0.9433 *2	0.867 *1

*1: *p*-value was evaluated by one-way analysis of variance; *2: *p*-value was evaluated by Fisher exact test; Abbreviations: F: female, M: male.

**Table 3 diagnostics-11-00039-t003:** Best-corrected visual acuity (BCVA), max intraocular pressure (Max IOP) and visual field defect (Aulhorn-Greve classification) in each Group.

Group (*n*)	BCVA	Max IOP (mmHg)	A-G
Group I (24)	−0.33 ± 0.74	30.13 ± 13.67	1.71
Group II (19)	−0.56 ± 0.74	27.21 ± 10.61	1.17 (18) *
Group III (14)	−0.07 ± 0.72	33.64 ± 15.14	2.57
*p*-value	0.179	0.386	0.202

A-G: Aulhorn-Greve; *p*-value was evaluated by one-way analysis of variance; * One case was missing.

**Table 4 diagnostics-11-00039-t004:** Correlation of the max IOP (mmHg) with anterior segment manifestations and vitreous findings at the presentation.

	KP (*n*)	Nodule in the Angle (*n*)	PAS (*n*)	Vitreous (*n*)
Positive	32.70 (30)	31.50 (44)	30.97 (35)	31.63 (16)
Negative	27.04 (27)	25.00 (13)	28.50 (22)	29.39 (41)
*p*-value	0.1005	0.02696	0.5183	0.5311

*p*-value was evaluated by one-way analysis of variance; Abbreviations: KP: Mutton-fat like keratic precipitates. PAS: peripheral anterior synechia.

**Table 5 diagnostics-11-00039-t005:** (**A**) Logistic regression analysis of VFD in Aulhorn-Greve (A–G) among the different parameters (i.e., patient age, gender, number of manifestations in the three different zones, KPs, nodule in the angle, and snowball opacity) as evaluated by the Hosmer-Lemeshow test; (**B**) Logistic regression analysis of glaucoma surgery among the different parameters (i.e., patient age, gender, number of manifestations in the three different zones, KPs, nodule in the angle, and snowball opacity) as evaluated by the Hosmer-Lemeshow test.

**(A)**		
**Expl Var**	**CE**	***p*** **-Value**
Intercept	−6.19	0.0982
Age	−0.05	0.3403
Gender	2.13	0.2164
Zone I	−0.03	0.9609
Zone II	−1.35	0.2712
Zone III	−0.92	0.3348
Max IOP	0.35	0.0151
KP	−0.72	0.6159
Nodule *	0.00	0.9982
SB	1.05	0.5370
**(B)**		
**Expl Var**	**CE**	***p*** **-Value**
Intercept	−6.19	0.0982
Age	−0.05	0.3403
Gender	2.13	0.2164
Zone I	−0.01	0.9887
Zone II	−1.98	0.0611
Zone III	1.96	0.0484
Max IOP	0.12	0.1330
A-G	0.99	0.0349
KP	0.31	0.8125
Nodule *	−2.30	0.3752
SB	1.07	0.5104

Abbreviations: Expl Var: explanatory variable; CE: coefficient; KP: existence of mutton-fat like keratic precipitates; Max IOP: maximum intraocular pressure; Nodule: existence of nodule in the angle; Zones I, II, and III: the number of abnormalities in fundus fluorescein angiography in the different zones (i.e., I, II, and III); SB: snowball opacity in vitreous; * Nodule: nodule in the angle.

## Data Availability

Data can be used by citing this article.

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
