# Peer review of "Microangiopathy in Ocular Sarcoidosis Using Fluorescein Gonio and Fundus Angiography from Diagnostic and Therapeutic Aspects"

_diagnostics, 2020, doi:10.3390/diagnostics11010039_

Round 1

Reviewer 1 Report

I reviewed the paper entitled “Microangiopathy in Ocular Sarcoidosis using 3 Fluorescein Gonio and Fundus Angiography from 4 Diagnostic and Therapeutic Aspects” by Teruhiko Hamanaka et al.  Authors In a retrospective study investigated vascular abnormalities in sarcoidosis using 22 fluorescein gonioangiography (FGA) to detect angle neovascularization (ANV), fundus fluorescein 23 angiography (FFA), and pathological specimens from the aspects of microangiopathy. In 57 24 sarcoidosis patients, clinical data were reviewed by dividing the cases into three.  They found the granulomas adjacent to blood vessels, including the Schlemm’s canal, and thickening of 32 the retinal blood vessel wall caused occlusion of those vessels. Photocoagulation was required for 33 retinal tears (14%) and the retinal blood vessel occlusion (7%). Suppression of IOP elevation via 34 continuous topical betamethasone may be important to avoid irreversible outflow-route changes 35 and optic-nerve damage, and the concept of microangiopathy in ocular sarcoidosis maybe 36 important for understanding the proper treatment of serious complications.  The review of the cases, illustrations are excellent.  The Paper has several interesting information that deserves to be published

Author Response

Thank you very much for reviewing our manuscript. We sincerely appreciate for your acceptance for publication in Vision.

Teruhiko Hamanaka

Reviewer 2 Report

Hamanaka et al. described the presence of microangiopathy and vascular abnormalities in ocular sarcoidosis (OS) by using an integration between different in vivo and ex vivo imaging technologies. I think that this paper could be a real masterpiece in diagnostic field of ocular sarcoidosis as it solves past diagnostic difficulties. Further, the work is original and the long time of clinical observations, according a really accurated clinical protocol, contributes to validate methods used for OS diagnosis. In vivo images of both FFA and FGA, integrated with ex vivo hystochemical and Electron microscopy analyses, give a significant contribution to analyze microscopic vascular abnormalities, important for early diagnosis of this pathology.

I have only few suggestions to make the manuscript even more understandable:

1- English language is almost perfect but I noticed some typos which you can correct easily.

2- Improve the quality of FFA and FGA images and explain better the results integration between in vivo imaging and ex vivo analyses.

3- In methods section, describe better FFA and FGA imaging methods (in which way they were performed?) to make them understandable and reproducible by other interested clinicians.

Kind regards

Author Response

Thank you very much for reviewing our abstract. The detail of the method of slit lamp FGA and funds camera FGA were written in the manuscript (new reference number 14). According to the reviewer comment, the reference number 14 and 15 has changed and parts written with red letters in the line numbers 88-89, 100, 125, 127, 137, 939, 1072, 1073, 1184-1190 were changed. In the Figure 9B inset, visibility of retinal tear may have been increased. In Figure 8B the number of arrow heads was reduced showing the beginning area of avascular zone.

We sincerely appreciated your comments.